# Impact of Long-Term Pyriproxyfen Exposure on the Genetic Structure and Diversity of *Aedes aegypti* and *Aedes albopictus* in Manaus, Amazonas, Brazil

**DOI:** 10.3390/genes15081046

**Published:** 2024-08-08

**Authors:** Lorena Ferreira de Oliveira Leles, Marcus Vinícius Niz Alvarez, Jose Joaquin Carvajal Cortes, Diego Peres Alonso, Paulo Eduardo Martins Ribolla, Sérgio Luiz Bessa Luz

**Affiliations:** 1Laboratório de Ecologia de Doenças Transmissíveis na Amazônia, Instituto Leônidas e Maria Deane—Fiocruz Amazônia, Manaus 69027-070, Brazil; lorenaleless@gmail.com (L.F.d.O.L.); jjcarvajalc166@gmail.com (J.J.C.C.); sergio.luz@fiocruz.br (S.L.B.L.); 2Programa de Pós-Graduação em Biologia Parasitária, Instituto Oswaldo Cruz (IOC), Rio de Janeiro 21040-900, Brazil; 3Laboratório de Pesquisa em Análises Genéticas, Instituto de Biotecnologia e Biociências, Universidade do Estado de São Paulo (UNESP), Botucatu 18607-440, Brazil; marcus.alvarez@unesp.br (M.V.N.A.); diego.p.alonso@unesp.br (D.P.A.)

**Keywords:** *Aedes aegypti*, *Aedes albopictus*, pyriproxyfen, ddRADseq, genomic surveillance

## Abstract

*Aedes aegypti* and *Aedes albopictus* are responsible for transmitting major human arboviruses such as Dengue, Zika, and Chikungunya, posing a global threat to public health. The lack of etiological treatments and efficient vaccines makes vector control strategies essential for reducing vector population density and interrupting the pathogen transmission cycle. This study evaluated the impact of long-term pyriproxyfen exposure on the genetic structure and diversity of *Ae. aegypti* and *Ae. albopictus* mosquito populations. The study was conducted in Manaus, Amazonas, Brazil, where pyriproxyfen dissemination stations have been monitored since 2014 up to the present day. Double digest restriction-site associated DNA sequencing was performed, revealing that despite significant local population reductions by dissemination stations with pyriproxyfen in various locations in Brazil, focal intervention has no significant impact on the population stratification of these vectors in urban scenarios. The genetic structuring level of *Ae. aegypti* suggests it is more stratified and directly affected by pyriproxyfen intervention, while for *Ae. albopictus* exhibits a more homogeneous and less structured population. The results suggest that although slight differences are observed among mosquito subpopulations, intervention focused on neighborhoods in a capital city is not efficient in terms of genetic structuring, indicating that larger-scale pyriproxyfen interventions should be considered for more effective urban mosquito control.

## 1. Introduction

*Ae. aegypti* (Linnaeus, 1762) and *Ae. albopictus* (Skuse, 1894) are among the most important mosquito species, responsible for transmitting major human arboviruses such as Dengue, Zika, and Chikungunya, which makes them a global threat to public health [1,2]. Due to the lack of etiological treatments and efficient vaccines for those arboviruses, except for yellow fever and recently Dengue, vector control strategies have become essential for reducing vector population density, interrupting the pathogen transmission cycle, and reducing risks of epidemics [3,4].

*Aedes* spp. vectors are frequent targets of government and World Health Organization control campaigns. Although approximately 60% of investments in vector control between 2018 and 2030 will be in the development of new strategies and insecticides, current strategies still consist of adulticides in households and the identification and elimination of breeding sites by health agents [5]. Even with the benefits of these strategies, such as a drastic reduction in vector populations, the coverage rate of treated breeding sites is low due to the biology of these vectors that skip-oviposit in inaccessible breeding sites [6].

A promising control strategy uses adult mosquitoes and dissemination stations (DSs) to self-disseminate potent larvicides, such as pyriproxyfen (PPF) [6,7]. This strategy aims to increase the rate of treated breeding sites, which impairs the life cycle of *Aedes* spp. and ultimately impacts arbovirus transmission [7,8,9]. Despite the effectiveness of this larvicide and its recommended use in Brazil by the National Dengue Control Program (PNCD), there are few studies on the role of this control strategy on the structure and population genetics of these vectors [4,10].

The impact of the control strategy with PPF larvicide DSs remains unknown in terms of the biological dynamics, genetic, and population structure of vectors of *Aedes* spp. [8,11]. For *Ae. aegypti* and *Ae. albopictus*, studies that correlate population genetics structure under the influence of insecticide use are scarce and exclusively based on primer-specific SNP (single nucleotide polymorphism) genotyping and microsatellite markers [12,13,14].

Given the development of new DNA sequencing techniques and more robust molecular analyses, the scope of data obtained allows a more reliable understanding of consequent genetic stratification of vector populations based on the detection of significant microvariations or single nucleotide polymorphism (SNP) [10,15]. Such data could be informative on how these insecticides may influence the population dynamics, not only of *Aedes* spp. but also for other vectors relevant to public health [3,10,12]. Additionally, despite the large genome size of *Ae. aegypti* and *Ae. albopictus*, double digest restriction-site associated DNA sequencing (ddRADSeq) effectively recovers nuclear genome markers, allowing us to obtain robust genomic data associated with this highly reliable technique while significantly reducing the per base sequencing cost [16,17,18].

In this study, we applied a high-throughput sequencing method (ddRADSeq) to analyze the impact of long-term pyriproxyfen exposure by dissemination stations on the genetic structure and diversity of *Ae. aegypti* and *Ae. albopictus* mosquito populations in monitored areas of Manaus, the capital of the state of Amazonas, Brazil. This approach allows us to assess the effects of PPF larvicide over a spatiotemporal period, comparing control and experimental sites in an urban area. Additionally, this sequencing approach, which allows us to obtain sufficient genomic data with high reliability for understanding population genetic structure, can also contribute to activities such as monitoring mosquito population migrations as well as their geographical dispersion, identifying genes associated with insecticide resistance for development and implementation of effective control strategies, evaluating the efficacy of control strategies such as the release of genetically modified mosquitoes or implementation of insecticide-based control programs, and understanding the dynamics of vector populations in order to predict and interrupt the transmission of arboviruses, thus improving genomic and epidemiological surveillance.

## 2. Materials and Methods

### 2.1. Study Area and Mosquito Surveillance

Samples for this study were collected in Manaus, Amazonas, Brazil, a city covering approximately 11,401,092 km^2^, with 2,063,689 inhabitants and 179,288 households (according to data from the Brazilian Institute of Geography and Statistics—IBGE/2022). Control entomological samples were collected in the neighborhoods of Adrianópolis (−3.09447, −60.00744) and Aleixo (−3.08840, −59.99265)(AdAl), starting in 2017 and still ongoing. Experimental entomological samples were collected in the Glória neighborhood (−3.11948, −60.03449) during the same period (Figure 1). Fifty collection points were randomly selected to set PPF DSs across the neighborhood to provide maximum coverage of the control and experimental sites. DSs were verified twice a month from August 2017 until the present year, with a pause in 2020 (April–May) and 2021 (January–February) due to the COVID-19 pandemic. The juvenile and adult specimens of *Ae. aegypti* and *Ae. albopictus* were collected and morphologically identified. The collected samples were stored in 100% alcohol and placed in a fridge (4 °C).

The dissemination station (DS) is a cylindrical plastic container with 1.5 L of water and black fabric impregnated with 0.5% micronized pyriproxyfen. Once *Aedes* females get onto the tissue, they become impregnated with larvicide and act as larvicide disseminators in other breeding sites. In the Glória neighborhood, 316 DSs were installed, and for evaluation, egg collections were carried out using ovitraps, and egg entomological indexes were calculated monthly. Over 46 months, the positive ovitrap index (POI) in the Glória neighborhood was 53.6% (CI 47.5–59.7), while in the AdAl neighborhood was 71.2% (CI 66.8–75.6). The egg mean index (EMI) in the Glória neighborhood was 18.8 eggs/ovitrap (CI 15.9–21.7), and in the AdAl neighborhood was 28.7 eggs/ovitraps (CI 24.7–32.7).

### 2.2. Sample Preparation and Sequencing

For DNA extraction, each specimen was extracted individually using the DNeasy Blood & Tissue kit (Qiagen, Germantown, MD, USA). DNA concentration and purity were assessed by fluorometric quantitation using the QuBit dsDNA HS Assay Kit (Thermo Fisher Scientific Inc. Waltham, MA, USA). Both techniques were performed according to the manufacturer’s recommendations.

DNA libraries were prepared, and samples were sequenced by double digest restriction-site associated DNA technique in a 151-cycle single-read run and using EcoRI-MspI restriction enzymes, as described in Campos et al., 2017. Sequencing quality control analyses were performed using the FASTQC program [19] before and after read filtering.

### 2.3. Species Identification

Sequencing data were aligned to *Ae. aegypti* and *Ae. albopictus* cytochrome oxidase subunit I (COI) reference sequence (available at KC913582.1 and NC_006817.1) using Burrows–Wheeler Aligner (BWA) software v1.20 [20]. After alignment, individual COI consensus sequences were generated using the SamTools software package v0.7.17 [21]. BLASTn tool was used for multi-species identification using the individual COI consensus sequences [22]. Only the highest matching result from BLAST was used. Specimens were discarded if e-value > 1 × 10^−100^, identities < 200, identity < 90%, and the matching sequence was not identified as one of the species.

### 2.4. Variant Calling

Sequencing quality control was applied with mean quality filtering, trimming, minimum length filtering (80 bp), and adapter removal procedures using Trimmomatic [23]. All sequencing reads were aligned to the reference genome of *Ae. aegypti* (NC_035159.1) and *Ae. albopictus* (JAFDOQ000000000.1), both retrieved from the NCBI database by the BWA program [20]. Variant calling was performed using Stacks v2.62 software [21] with gstacks approach. The variant panel was exported in the VCF v4.2 format. SNPs were removed from the variant panel based on a minimum allele frequency (MAF) < 0.1 and at least 8 non-missing genotypes within each group.

### 2.5. Population Genetics Structure Inferences

Statistical analyses were performed with PLINK 1.9 [24], and graphs and figures were generated using the GGPlot2 version 3.5.0 [25] for R in RStudio [26]. Pairwise F_ST_ was estimated to check population differentiation with Arlequin 3.5 [27]. The principal component analysis (PCA) was calculated based on pairwise genetic distance IBS (identity-by-state matrix), while hierarchical clustering analysis was performed using pvclust v2.2-0 package [28] for R [29].

PERMDISP version 2.6-4 analysis [30] was performed using a vegan package for R to evaluate whether there were significant differences in variability among groups were significant. Nucleotide diversity was estimated for each group to address changes in genetic variability. Median Tajima’s D was calculated with VCFtools v0.1.16 [31] for each group to estimate deviations from neutrally, and the Man–Whitney test was applied to check significant differences.

## 3. Results

### 3.1. Data Collection and DNA Extraction

During the monthly checks from 2017 to 2020, which were interrupted due to the COVID-19 pandemic, 55,472 specimens of *Ae. aegypti* and 19,720 specimens of *Ae. albopictus* among juveniles and adults were collected. A total of 102 individuals were submitted to double digest restriction-site associated DNA sequencing (ddRADSeq), followed by species identification: 52 individuals from the control neighborhood, where PPF was not applied, and 50 individuals from the experimental neighborhood, considering both vector species (Table 1).

The initial approach would involve selecting 500 individuals, considering 100 collection sites combined from control and experimental groups, in a spatiotemporal experimental design between 2017 and 2020. However, the costs associated with sequencing are prohibitive, making it unfeasible to sequence this number of samples. Despite this limitation, we selected samples from the same collection points, considering the annual and rainy seasonal period when there is a higher availability of vectors, trying to be as geographically representative as possible, even though we are aware that this sample size may not fully represent the study areas across different sites and years.

It is important to emphasize that since PPF intervention was conducted in a scenario of neighborhoods located in a capital city, combined with the control site that had not been exposed to the intervention, the number of specimens collected was very large. Based on the mentioned collection points, the specimens selected for DNA extraction and analyses were made considering both sequencing cost and adequately representing both study areas (control and experimental).

### 3.2. Double Digest Restriction-Site Associated DNA Sequencing (ddRADseq) Performance

For *Ae. aegypti*, a total of 4,525,210 reads were properly mapped to the reference genome. The average sequencing depth was approximately 0.0032 ± 0.0583. After variant calling, the final dataset consisted of 1443 SNPs distributed throughout the genome. The SNP average sequencing depth and quality (Phred score) were 1.64 ± 1.79 and 23.11 ± 4.69, respectively. The SNP density was 0.00121 SNP/Kbp.

For *Ae. albopictus,* a total of 3,522,816 reads were properly mapped to the reference genome. The average sequencing depth was approximately 0.0026 ± 0.0037. After variant calling, the final dataset consisted of 645 SNPs distributed throughout the genome. The SNP average sequencing depth and quality (Phred score) were 1.65 ± 1.29 and 23.12 ± 4.95, respectively. The SNP density was 0.00125 SNP/Kbp.

### 3.3. Statistical and Structural Analysis

For *Ae. aegypti*, significant Theta(Pi) molecular diversity estimates were observed for control and experimental groups (0.4 and 0.369, respectively). The estimates of Tajima’s D of control and experimental groups were, respectively, 1.41 and 1.16 (*p*-value < 0.00454). Based on the F_IT_ estimate, no significant differences were observed for homozygosity (F~0.84, *p*-value < 0.87).

The median pairwise IBS distance between samples within groups showed significant differences between control and experimental groups, 0.109 and 0.0631 (*p*-value < 2.2 × 10^−16^), respectively. Significant stratification was observed between control and experimental groups, with a pairwise F_ST_ estimate of 0.015 (*p*-value < 0.037, number of permutations = 1000). Significant stratified sites were observed based on 70 SNPs (number of permutations = 1000).

PCA analysis showed SNP clustering for control and experimental groups (Figure 2A). The clustering is mainly represented by the first principal component, which corresponds to the major explained variance (19.8%). PERMDISP analysis revealed significantly different variability between groups [F(1.50) = 6.05, *p* = 0.02*]. The association test between control and experimental groups presented two significant SNPs (Figure 2B) adjacent to the gene region of the “neuropeptide CCHamide-2 receptor”. The SNPs were found at the positions (NC_035109.1:96061807:A:G) e (NC_035109.1:96061808:A:G).

For *Ae. albopictus*, Theta(Pi) molecular diversity estimates were 0.366 for both groups, showing that there is no significant molecular diversity between the control and experimental groups. The estimates of Tajima’s D of control and experimental groups were 1.01 and 1.03 (*p*-value < 0.3518). Based on the F_IT_ estimate, significant differences were observed for homozygosity (control = 0.794, experimental = 0.824, *p*-value = 0.049). The median pairwise IBS distance between samples within groups was 0.113 for both groups, showing no significant difference between control and experimental groups (*p*-value = 0.4777). No significant stratification was observed between the control and experimental groups, with a pairwise F_ST_ estimate of 0.0013 (*p*-value < 0.409, number of permutations = 1000).

PCA analysis showed clustering for the SNP genotype matrix, as shown in the figure below (Figure 3). The clustering is mainly represented by the first principal component which corresponds to the major explained variance by the model (21%). PERMDISP analysis revealed no significant variability between groups [F(1.50) = 0.07, *p* = 0.79].

## 4. Discussion

When we assessed the molecular diversity based on SNP typing for *Ae. aegypti*, a slight but significant difference was observed between the control and experimental groups, with approximately 7.8% nucleotide variation. This diversity was greater in the control group, which can be attributed to the absence of PPF intervention, which in turn could have resulted in no selective pressure on the local population and also a continuous migration flow from adjacent populations. It is noteworthy that the locations in this study correspond to neighborhoods in Manaus, making PPF intervention a local and/or focal strategy. A statistically significant difference was also observed between groups based on Tajima’s D estimate, indicating that *Ae. aegypti* populations in both localities are in a state of equilibrium/neutrality, corroborating nucleotide diversity data.

On the other hand, no significant differences in homozygosity were observed comparing individuals across the entire population (F_it_ estimate). IBS estimate within groups was statistically significant, revealing that in the control group locality, there is a smaller genetic distance, as well as a greater genetic divergence between individuals from two localities. The average F_ST_ value for *Ae. aegypti* was significant and considerably high, even among populations sharing the same urban space. Population genetic studies with *Anopheles darlingi*, another important vector for public health, present experimental designs on a microgeographic scale and pairwise F_ST_ values similar to those observed here. It is important to emphasize that the F_ST_ values described were calculated using all markers recovered from sequencing, along with a permutation test (1000 permutations) to verify the statistical significance of the average F_ST_ per SNP [15,32,33]. Additionally, the detection of 70 highly stratified and informative SNPs for clustering highlights a discrete pattern of stratification when comparing control and experimental groups. Despite the migratory flow of populations adjacent to these two groups, it is likely that distinct subpopulations exist in the analyzed localities. This finding highlights the possibility that factors such as pesticide use may be contributing to this differentiation.

PCA analysis (Figure 2A) demonstrated discrete clusters between control and experimental groups, but the most interesting datum observed was the difference in the dispersion of individuals within each group. The control group (AdAl) showed lower genetic distance among *Ae. aegypti* individuals compared to the experimental group (Glória), suggesting greater variability among individuals in the intervention site. The closer the points, the greater genetic similarity between individuals was observed, suggesting genetic similarity and reduced genetic diversity within the experimental group. By analyzing dispersion, the PC1 axis showed greater significance, explaining 19.8% of observed variations between the two groups. Although there is clustering in the control population, individuals are more dispersed, suggesting higher diversity. This finding corroborates the data from F_ST_ analysis, which, despite not showing a clear and abrupt clustering pattern, allows the identification of two groups with distinct nucleotide diversity. Our results indicate that, although there is no significant differentiation by geographic location, we observed a reduction in genetic variability due to the intervention, which is interestingly highlighted by genetic variability rather than genetic structure or composition. In other words, the intervention appears to affect genetic diversity by reducing variability, but it did not alter the genetic composition of the population.

The higher diversity of the control group can be attributed to two possible factors: (1) the control group has no contact with the PPF intervention, resulting in a high and constant population density between this locality and adjacent populations. Thus, even with migration flow, there is a balance in population density between control and neighboring populations; and (2) in the experimental group where the intervention occurs, several individuals are locally eliminated while adjacent populations continue to migrate constantly to this site, causing a significantly greater impact on population dynamics with a reduction of the population. Therefore, selective pressure influences the intervention site, leading to a reduction in genetic variation.

The association test based on allelic frequency (Figure 2B) revealed that among 70 highly stratified SNPs, two SNPs are clearly highlighted, showing different fixation levels being found exclusively in individuals from the control population (AdAl). Those SNPs are not located inside gene regions but rather in a region adjacent to neuropeptide receptor CCHamide-2. Recent studies have shown that this neuropeptide plays a fundamental role in the endocrine system of various arthropod species, such as *Drosophila melanogaster*, *Diaphorina citri*, and *Acyrthosiphon pisum* [34,35,36,37]. CCHa2-R allows endocrine cells in the midgut and peripheral adipose tissue to communicate with the central nervous system to ensure regulation of feeding, insulin production, lipid metabolism, development and growth, energy maintenance, and formation of appetitive and associative odor–sugar memory [38,39,40].

It is important to emphasize that although there are currently no studies of CCHa2-R neuropeptide in *Aedes* spp. mosquitoes, but given its function, the long-term exposure of these vectors to insecticides may influence the regulation of physiological processes related to sugar intake, as well as metabolic alterations that affect insecticide efficacy. Further investigations are necessary, which may suggest the development of control strategies using this receptor as a potential target. These data indicate that intervention significantly impacts population dynamics, leading to distinct subpopulations within both control and experimental groups of *Ae. aegypti*. Long-term exposure to PPF by the experimental group results in a bottleneck, highlighted by lower nucleotide diversity and higher average genetic distance between individuals, suggesting a drastic reduction in population density and incorporation of new individuals from adjacent populations. Additionally, PCA analysis suggests that the migration of new individuals into these locations contributes to different levels of diversity without changing the overall population’s genetic neutrality. Although the control and experimental groups are not entirely distinct in terms of stratification, they represent populations with different and significant parameters of diversity and genetic dispersion, reflecting the effects observed in PPF intervention.

When analyzing *Ae. albopictus* populations, statistically significant differences were observed based on homozygosity for F statistics (F_it_ estimate). Despite being subtle, these differences indicate that the level of homozygosity is higher in individuals from the experimental group, suggesting that they are genetically similar (Figure 3). By comparing the two species, the level of homozygosity between *Ae. aegypti* and *Ae. albopictus* was quite similar. However, the average value of the fixation index (F_ST_ estimate) was 10 times lower for *Ae. albopictus*, these data were not statistically significant in the permutation test. No stratification between the two locations was observed, indicating that individuals in both groups are genetically similar regardless of the presence of PPF, suggesting the occurrence of a single population and characterizing a scenario of panmixia.

No significant differences were observed for *Ae. albopictus* based on molecular diversity, neutrality tests, and IBS estimates within the same group, regardless of intervention status. Stratified SNP analyses and association tests were not conducted, as no stratification was observed between the control and intervention populations. It is noteworthy that continuous migration flow from adjacent populations also occurs, as experimental design and study sites correspond to those analyzed for *Ae. aegypti*.

The scenario observed for *Ae. albopictus* contrasts with the population dynamics observed for *Ae. aegypti*. The control and experimental populations showed minor differences based on the analyzed genetic parameters, indicating a homogeneous population in both locations. These results reflect the biology of the vector itself, as *Ae. albopictus* is less domesticated and anthropophilic and, unlike *Ae. aegypti*, does not depend on high urban concentration to thrive [41,42]. Although being found in urban and peri-urban environments, the mosquito commonly inhabits forested and frontier areas, vegetation, and plantations, primarily colonizing rural and wild environments, indicating a less defined population structure in urban settings compared to *Ae. aegypti* [1,43,44]. Another important characteristic is flight dispersal: while *Ae. aegypti* has more limited flight dispersal (range of 100–200 m), *Ae. albopictus* has greater dispersal capacity, tending to fly longer distances (between 400 and 600 m), which contributes to its geographic spread [45,46,47].

The results obtained so far indicate that focal intervention has a small or no impact on *Ae. albopictus* population found in urban areas in Manaus. Additionally, there are few studies regarding the population dynamics of this vector in Brazil due to its low importance as a local vector of arboviruses. Overall, focal intervention with PPF locally reduces the abundance of both vectors; however, in highly urbanized areas such as Manaus, vector population density is much higher, with a continuous migration flow among neighborhoods. Additionally, despite promoting a drastically reduced abundance of juvenile and adult mosquitoes, this intervention model presents challenges. As the selected neighborhoods designated as control and experimental groups are subjected to long-term intervention (since 2014) and, at the same time, are separated by a short geographical distance (4256 km apart in a straight line), observing significant genetic clustering on a microgeographic scale might be cumbersome.

Nevertheless, the population structure of *Ae. aegypti* correlates with its domestication and urbanization capacity, suggesting that it is more genetically structured and directly affected by PPF intervention. However, *Ae. albopictus* populations colonize urban areas less frequently, which results in more homogeneous and less structured populations. Thus, PPF intervention in this model may not have the expected effect on genetic structure besides interfering with clearer insights into the population dynamics of these vectors, as focal intervention does not affect neighboring populations that can rapidly colonize treated sites.

This study also provides valuable insights into the use of high-throughput sequencing, specifically ddRADSeq, to assess intervention strategies, including the impact of long-term pyriproxyfen exposure on the genetic structure and diversity of *Ae. aegypti* and *Ae. albopictus* populations. The data obtained highlight reductions in population density and arbovirus transmission, as well as changes in genetic structuring and dynamics, being crucial for understanding the genetic diversity of mosquito populations, allowing for the planning and development of customized control strategies for different regions and populations, thus increasing the effectiveness of interventions.

Additionally, these data can assist in other activities, such as understanding how the genetic structure of mosquitoes can influence their vector competence, identifying which mosquito populations are more likely to transmit arboviruses, developing specific diagnostic tools based on the identification of genetic markers that can quickly detect mosquito populations at high risk of arbovirus transmission, monitoring mosquito population migrations and their geographical dispersion, identifying genes associated with insecticide resistance for development and implementation of insecticide-based control strategies and programs, and understanding the dynamics of vector populations to predict and interrupt arboviruses transmission improving genomic and epidemiological surveillance. Despite the challenges and need for further studies, large-scale genomic surveillance using ddRADSeq shows promise for understanding the genetic structure and population dynamics of medically important vectors. Future investigations should explore migration patterns, larger geographic interventions, and the effects of physical barriers, as this is the first study to associate *Aedes* spp. with PPF intervention on a microgeographic scale in an extensive urban area.

## 5. Conclusions

Our results indicate that focal intervention application has no significant impact on the population stratification of *Ae. aegypti* and *Ae. albopictus* in urban scenarios. However, the level of structuring in *Ae. aegypti* suggests that this species is more genetically stratified and directly affected by PPF intervention. In contrast, the population of *Ae. albopictus*, being more homogeneous and less structured, did not show significant changes in terms of population genetics with a focal intervention strategy. However, slight differences are observed among *Ae. aegypti* subpopulations, it becomes clear that PPF intervention, based on this model focused on neighborhoods in a capital city, was not as efficient as expected in terms of structuring. This study provides valuable insights into the use of ddRADSeq to assess the impact of long-term PPF exposure on the genetic structure and diversity of *Ae. aegypti* and *Ae. albopictus* populations. This technique has proven to be a valuable tool for genetically characterizing diverse vector populations in association with intervention periods and approaches, contributing significantly to genomic surveillance. To date, this is the first study to evaluate the genetic structure of field-collected *Aedes* spp. exposed to intervention, and further studies are needed to better understand the structuring of vector populations associated with genomic surveillance, especially for *Aedes* spp.

## Figures and Tables

**Figure 1 genes-15-01046-f001:**
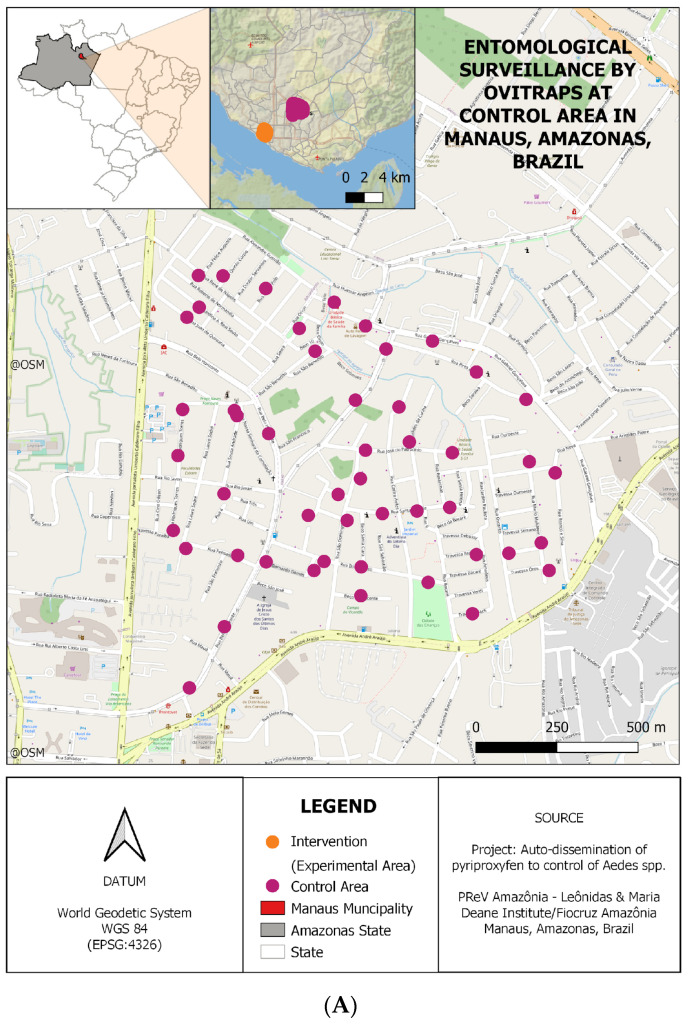
Study site: capital city of Manaus, Amazonas, Brazil. (**A**) Representation of collection points in AdAl neighborhood (control) and (**B**) Glória neighborhood (experimental). The dots indicate the location of the ovitraps monitored for mosquito vectors. The map was created using the maps library on the software QGIS (v.3.22.14—Białowieża).

**Figure 2 genes-15-01046-f002:**
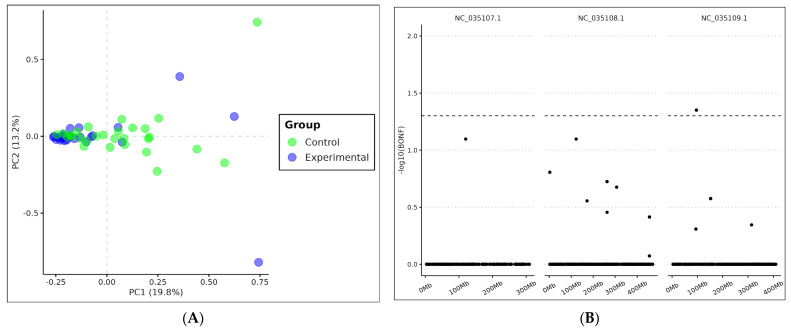
(**A**) Principal component analysis biplot for *Ae. aegypti* based on genotype matrix. Values between parentheses represent the variance explained by the respective principal component. (**B**) Manhattan plot of case/control allelic test (1Df Χ^2^). BONF—Bonferroni adjusted *p*-value. Points above the dashed line were considered statistically significant.

**Figure 3 genes-15-01046-f003:**
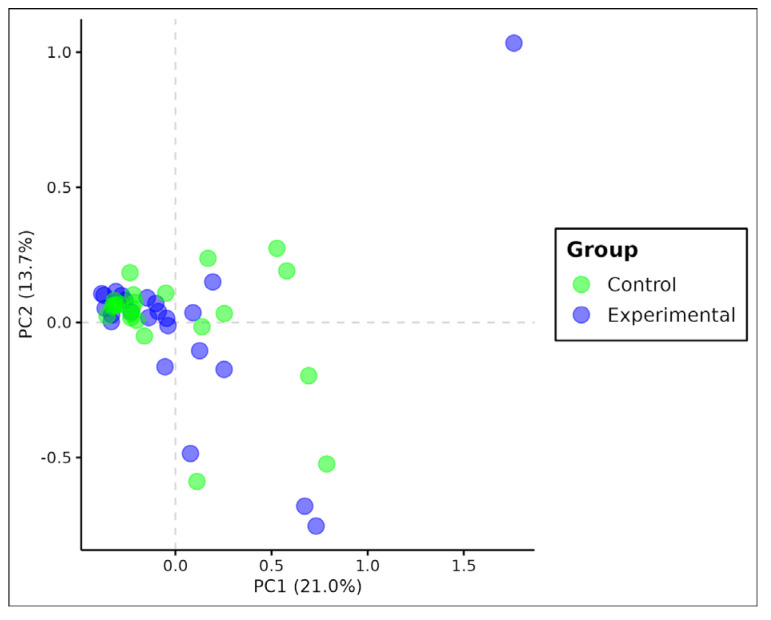
Principal component analysis biplot for *Ae. albopictus* based on genotype matrix. Values between parentheses represent the variance explained by the respective principal component.

**Table 1 genes-15-01046-t001:** Individuals were selected from each species for ddRAD sequencing.

	*Ae. aegypti*		*Ae. albopictus*	
AdAl—Control	Total	Selected	Total	Selected
JAN/FEB—2018 to 2020	28,311	28	16,820	23
Glória—Experimental	Total	Selected	Total	Selected
JAN/FEB—2018 to 2020	27,161	25	2900	25

## Data Availability

Data are available at ZENODO with the following number: 12636870 (available online at https://doi.org/10.5281/zenodo.12636870).

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
