# Peer review of "Impact of Long-Term Pyriproxyfen Exposure on the Genetic Structure and Diversity of Aedes aegypti and Aedes albopictus in Manaus, Amazonas, Brazil"

_genes, 2024, doi:10.3390/genes15081046_

Round 1

Reviewer 1 Report

Comments and Suggestions for Authors

This study investigates the impact of long-term pyriproxyfen exposure on the genetic structure and diversity of Ae. aegypti and Ae. albopictus mosquito populations. By monitoring mosquito populations at local dissemination stations since 2014 and using digest restriction-site associated DNA sequencing techniques, the authors report no significant stratification of Aedes aegypti and Ae. albopictus populations at their experimental sites compared to the control sites. Overall, the authors' results have implications for our understanding of the long-term impact of insecticides on the dynamics of mosquito population genetics. However, several elements of the manuscript are difficult to interpret in their current form.

Major Issues

  1. Title: The current title is vague; a more specific title that precisely reflects the topic of the research is recommended.
  2. Abstract: The relevance of investigating the genetic structure and diversity of mosquitoes to vector control and pathogen transmission interruption is not clear. The indexes used to measure genetic structure and diversity should be clarified. Additional information in the abstract and materials and methods would improve clarity and readability.
  3. Introduction: A paragraph bridging the gap between vector control and mosquito genetic structure/diversity should be added. Additionally, the hypotheses/predictions and the experimental approaches/design used to test these hypotheses should be clearly stated in the last paragraph.
  4. Materials and Methods: More technical details are needed on how insecticides were applied to local mosquito populations through dissemination stations and how experimental and control mosquito populations were sampled. The rationale for using population stratification to represent genetic structure and diversity needs justification. More details on statistical analyses, including which models/analyses were used to address specific scientific questions/predictions and why, should be provided.
  5. Results: The total number of specimens is high (55,472 and 19,720), but it is unclear why only 102 were selected for DNA extraction and downstream analyses. The selection process for the 102 specimens should be clarified. The biological significance of significant or non-significant effects should be mentioned or prioritized. The results section needs to be rewritten to explain the findings in terms of the biological hypotheses rather than just describing model outputs.
  6. Discussion: The relevance of understanding genetic structure and diversity to vector control and vector-borne disease dynamics should be thoroughly discussed, which may enhance the significance and impact of this study.

Minor Issues

  1. L19-20: I don’t see any results of this manuscript supporting the statement “significant local population reductions”.
  2. L22: Provide the full name of the abbreviation when it first appears.
  3. L53: Species names are not correctly italicized; this issue occurs throughout the manuscript (e.g., L147, L168, L185, L193). Please correct them.
  4. L60: Provide the full name of the abbreviation when it first appears.
  5. L88: Is it "ovitraps" or "ovitramps"?
  6. L96-98: Provide the amplification primers and technical parameters for quality control.
  7. L105-107: Any references for using these criteria?
  8. Inconsistent number format: Some numbers use a comma (e.g., L127, L136) to separate every three digits, while others use a dot (e.g., numbers in Table 1, L141). Please use a consistent format to avoid confusion.
  9. Figures: the significant or non-significant effects should be clearly indicated in the figures or the legends.
Comments on the Quality of English Language

The quality of English language is good.

Author Response

Questions for General Evaluation

Response 1: Thank you for your general evaluation. We hope we have managed to improve the presentation of our methods and conclusions for greater clarity of our manuscript.

Point-by-point response to Comments and Suggestions for Authors

Comment 1: This study investigates the impact of long-term pyriproxyfen exposure on the genetic structure and diversity of Ae. aegypti and Ae. albopictus mosquito populations. By monitoring mosquito populations at local dissemination stations since 2014 and using digest restriction-site associated DNA sequencing techniques, the authors report no significant stratification of Aedes aegypti and Ae. albopictus populations at their experimental sites compared to the control sites. Overall, the authors' results have implications for our understanding of the long-term impact of insecticides on the dynamics of mosquito population genetics. However, several elements of the manuscript are difficult to interpret in their current form.

Response 1: Dear Reviewer,

We deeply appreciate your considerations regarding our manuscript. We value your suggestions and ensure that all raised concerns will be carefully addressed to improve the overall quality of the manuscript. We thank you for the time and effort dedicated to reviewing our work.

Major Issues

- Comment 2 – Title: The current title is vague; a more specific title that precisely reflects the topic of the research is recommended.

Response 2: Thank you for the suggestion. The title has been changed to "Impact of long-term pyriproxyfen exposure on the genetic structure and diversity of Aedes aegypti and Aedes albopictus in Manaus, Brazil" to better reflect the content of the manuscript.

Comment 3 – Abstract: The relevance of investigating the genetic structure and diversity of mosquitoes to vector control and pathogen transmission interruption is not clear. The indexes used to measure genetic structure and diversity should be clarified. Additional information in the abstract and materials and methods would improve clarity and readability.

Response 3: We agree and appreciate this suggestion. The abstract content has been modified for better clarity and readability. However, since the abstract can have a maximum of 200 words, this limits the addition of further information on this topic. Nevertheless, more technical details have been included in the materials and methods to improve the clarity of our manuscript.

Comment 4 – Introduction: A paragraph bridging the gap between vector control and mosquito genetic structure/diversity should be added. Additionally, the hypotheses/predictions and the experimental approaches/design used to test these hypotheses should be clearly stated in the last paragraph.

Response 4: We appreciate your suggestion to include a paragraph to provide a clearer context for our introduction. Additionally, we have updated the final paragraph to clearly state the hypotheses and the experimental approaches used to test them. This will help clarify our manuscript. Thank you for your feedback.

Comment 5 – Materials and Methods: More technical details are needed on how insecticides were applied to local mosquito populations through dissemination stations and how experimental and control mosquito populations were sampled. The rationale for using population stratification to represent genetic structure and diversity needs justification. More details on statistical analyses, including which models/analyses were used to address specific scientific questions/predictions and why, should be provided.

Response 5: Additional information regarding technical details on insecticide application and collection, population stratification, and analyses has been provided in the revised manuscript. Thank you for your considerations aimed at improving the understanding of our manuscript. We appreciate your suggestion.  

Comment 6 – Results: The total number of specimens is high (55,472 and 19,720), but it is unclear why only 102 were selected for DNA extraction and downstream analyses. The selection process for the 102 specimens should be clarified. The biological significance of significant or non-significant effects should be mentioned or prioritized. The results section needs to be rewritten to explain the findings in terms of the biological hypotheses rather than just describing model outputs.

Response 6: Thanks for your comment. We acknowledge that the total number of collected specimens is high, and we would like to clarify that the specimens selected for DNA extraction and analyses were chosen to adequately represent the study areas (control and experimental). This selection was made considering both sequencing cost and available financial resources, which may limit the number of samples that can be processed. In the revised manuscript, we have restructured the results section to explain them more clearly. We appreciate your observations once again and are available for further discussions.

Comment 7 – Discussion: The relevance of understanding genetic structure and diversity to vector control and vector-borne disease dynamics should be thoroughly discussed, which may enhance the significance and impact of this study.

Response 7: We appreciate your comment and agree on the importance of discussing the relevance of genetic structure and diversity in the context of vector control and vector-borne disease dynamics. In response to your suggestion, we have added a section to the manuscript discussion that addresses how understanding genetic structure and diversity can impact vector control strategies and influence the dynamics of vector-borne diseases. We hope that this addition will enhance the impact of our study, as suggested.

Minor Issues

Comment 8 – L19-20: I don’t see any results of this manuscript supporting the statement “significant local population reductions”.

Response 8: We agree with this comment. The abstract content has been modified for better clarity and readability. Our research group has been working with pyriproxyfen dissemination stations since 2014 in various locations across Brazil, included other neighborhood in Manaus. The data pertaining to the sites in this manuscript have not yet been published. However, other studies using this approach, such as Abad-Franch et al. 2015 (doi:10.1371/journal.pntd.0003702) and Abad-Franch et al. 2017 (doi:10.1371/journal.pmed.1002213), demonstrate not only significant reductions in populations at treatment sites but also cases where the populations reach nearly zero. Thanks for this observation.

Comment 9 – L22: Provide the full name of the abbreviation when it first appears.

–L53: Species names are not correctly italicized; this issue occurs throughout the manuscript (e.g., L147, L168, L185, L193). Please correct them.

– L60: Provide the full name of the abbreviation when it first appears. 

–L88: Is it "ovitraps" or "ovitramps"?

Response 9: The first mentions of a given term were corrected from the abbreviated form to the full written form, and the names of the species were checked and corrected in italics throughout the manuscript.

Comment 10 – L96-98: Provide the amplification primers and technical parameters for quality control.

Response 10: There are no amplification primers. We generated COI consensus using only the covered regions by the RadSeq and submitted the resulting subsequences to BLASTn.

Comment 11 – L105-107: Any references for using these criteria?

Response 11: No, these are arbitrary parameters for quality control. We used FastQC for sequencing quality reports and based on the report, we defined filtering steps and parameters based on available examples on Trimmomatic v0.32 manual.

Comment 12 – Inconsistent number format: Some numbers use a comma (e.g., L127, L136) to separate every three digits, while others use a dot (e.g., numbers in Table 1, L141). Please use a consistent format to avoid confusion.

Response 12: The number format has been corrected to avoid confusion: comma for separating thousands and period for separating decimals. Thank you for the valuable correction.

Comment 13 – Figures: the significant or non-significant effects should be clearly indicated in the figures or the legends.

Response 13: The legends have been modified to more clearly represent the effects considered statistically significant. Thank you.

Response to Comments on the Quality of English Language

Point 1: (x) Minor editing of English language required.

Response 1: Thank you for considering our manuscript well-written in English. All suggested corrections throughout the text have been checked. Thank you for your dedication in reviewing our manuscript.

Reviewer 2 Report

Comments and Suggestions for Authors

Review comments

In the manuscript, “Use of Genetics in Monitoring Control Measures for Aedes aegypti and Aedes albopictus,” the authors have chosen a highly relevant topic by investigating the use of ddRAD-seq in monitoring gene flows of Aedes aegypti and Aedes albopictus in cities of Brazil where pyriproxyfen is disseminated for controlling mosquito populations. This research addresses a significant gap in existing literature.

The manuscript is well-written and reads well. It provides sufficient information to comprehend in the introduction section, and the methods were well described. The results are clearly presented, and the discussions are congruent with the results. As a reviewer, I do not have any reservations about suggesting the manuscript with some minor cosmetic revisions.

Below are specific suggestions to correct cosmetic errors:

- Line 53, convert ‘Albopictus’ to ‘albopictus’

- Line 88, replace ‘ovitramps’ with ‘ovitraps’

- Line 168, italicize ‘Ae. albopictus’

- Ensure that species names in the results and discussion sections are consistently italicized to match the introduction and methods sections.

Author Response

Questions for General Evaluation

Response 1: Thank you for your general evaluation of our manuscript. We hope we have managed to improve the presentation of our results and conclusions for greater clarity of our paper.

Point-by-point response to Comments and Suggestions for Authors

Comment 1: In the manuscript, “Use of Genetics in Monitoring Control Measures for Aedes aegypti and Aedes albopictus,” the authors have chosen a highly relevant topic by investigating the use of ddRAD-seq in monitoring gene flows of Aedes aegypti and Aedes albopictus in cities of Brazil where pyriproxyfen is disseminated for controlling mosquito populations. This research addresses a significant gap in existing literature.

The manuscript is well-written and reads well. It provides sufficient information to comprehend in the introduction section, and the methods were well described. The results are clearly presented, and the discussions are congruent with the results. As a reviewer, I do not have any reservations about suggesting the manuscript with some minor cosmetic revisions.

Response 1: Dear Reviewer,

We deeply appreciate your comments and considerations regarding our manuscript. We are very pleased to know that you recognize the relevance and importance of our research in filling a significant gap in the current literature. We value your valuable suggestions and ensure that all raised points will be addressed to improve the quality of the manuscript.

Below are specific suggestions to correct cosmetic errors:

Comment 2: Line 53, convert ‘Albopictus’ to ‘albopictus’

Response 2: The correct spelling of the term has been corrected.

Comment 3: Line 88, replace ‘ovitramps’ with ‘ovitraps’

Response 3: The term has been replaced with the correct form.

Comment 4: Line 168, italicize ‘Ae. albopictus’

Response 4: The term has been correctly italicized.

Comments 5: Ensure that species names in the results and discussion sections are consistently italicized to match the introduction and methods sections.

Response 5: The entire text has been reviewed again to ensure that the species names throughout the manuscript are consistently italicized.

Response to Comments on the Quality of English Language

Point 1: (x) English language fine. No issues detected.

Response 1: Thank you for considering our manuscript written in English to be of high quality and for your dedication in reviewing our manuscript.

Reviewer 3 Report

Comments and Suggestions for Authors

The work by Ferreira de Oliveira Leles et al. deals with the detection of population structure in two mosquito species which transmit arboviruses in the city of Manaos. They followed the pesticide application during several years and investigated if this treatment had had any impact at the neighborhood level. The work is worthy because of the big problem that the dengue fever is currently around the globe. I list some concerns which, hava they been addressed, could improve the overall quality of the manuscript.

Line 53: Italicize "albopictus".

Lines 72-73: It seems the authors have used the decimal point in Portuguese style. Instead, use comma for separating thousands and stop for separating decimals.

Line 80: I think that the Ds abbreviation must be clarified in line 78, as follows, "Dissemination stations (DS)".

Line1: not in 2020?

Lines 100-107: It is not clear if the authors amplified a fragmento of the COI in orde to identify the mosquito species (if this is the case, please give the rpimers and PCR conditions). or if the authors search for mitochondrial SNPs in the reads with a program like mitoXXXX and then blaste these SNPs. Give a clearer explanation.

Line 113: delete the stop after [18].

Line 17: replace "Population and structure genetics inferences" by "Population genetics structure inference" or "Genetic structure inference".

Line 119: Provide the GGPlot2 package reference, as well as the R reference.

Line 123: Ib. pvclust v2.2-0 package.

Line 136: Italicize "Ae. aegypti".

Lines 137-145: It seems like very poor sequencing depth and quality for both species. I was expecting something values ca. 20-30X. The same for quality (Phred score), ca. 35-40

Line 141: Italicize "Ae. albopictus ".

Line 147: Italicize "Ae. aegypti".

Lines 150-151: I would rather expect estimations of both Fis (deviation from HW proportions) and Fst (population structure).

Line 155: Fst 0.015 does not indicate great differentiation. It would be interesting to estimate Fst within each neighborhood, to see is pesticide is a factor introducing (or erasing) signs of structuration. If data are pooled, the effect of pesticide cannot be distinguished.

 Italicize all scientific names across the whole manuscript and give the species authority.

Lines 192-194:  If both groups are in equilibrium, then how come that Tajima’s D are statistically different? Inf both are in equilibrium, then Tajima’s D for both groups do not differ statistically from 0. Thus, how both groups are different in this regard?

Lines 1999-200: Is an Fst = 0.015 large enough using this kind of markers? Can you provide comparisons with other systems?

Lines 205-207: I can see overlapping between both groups, although the Control group is far more diverse.

Lines 208-209: I think is the opposite pattern. Since the Control group is more diverse, then this should be the more variable group (I understand than the “intervention site” is the area of pesticide application, is that true?). The intervention (application of pesticide), must have exerted some selection for sure, lowering the genetic variation.

Lines 209-211: that is true. Maybe I misunderstood the term “intervention site”.

Lines 217-224: In other words, a selective pressure is acting in the intervention site, reducing genetic variation.

Comparison with other studies should be useful. I suggest at least try to run a software like Structire or Admixture, in order to better reflect the lack of structure. 

Author Response

Questions for General Evaluation

Response 1: Thank you for your general evaluation of our manuscript. We hope we have managed to improve our data presentation and conclusions for greater clarity of our paper.

Point-by-point response to Comments and Suggestions for Authors

Comment 1: The work by Ferreira de Oliveira Leles et al. deals with the detection of population structure in two mosquito species which transmit arboviruses in the city of Manaos. They followed the pesticide application during several years and investigated if this treatment had had any impact at the neighborhood level. The work is worthy because of the big problem that the dengue fever is currently around the globe. I list some concerns which, hava they been addressed, could improve the overall quality of the manuscript.

Response 1: Dear Reviewer,

We deeply appreciate your considerations regarding our manuscript. We are very pleased to know that you recognize the relevance of our work. We value your suggestions and ensure that all raised concerns will be carefully addressed to improve the overall quality of the manuscript. We thank you for the time and effort dedicated to reviewing our work.

Comment 2: Line 53: Italicize "albopictus".

Response 2: The correct spelling of the term has been corrected.

Comment 3: Lines 72-73: It seems the authors have used the decimal point in Portuguese style. Instead, use comma for separating thousands and stop for separating decimals.

Response 3: Corrections have been made throughout the manuscript to use commas (for separating thousands) and periods (for separating decimals).

Comment 4: Line 80: I think that the Ds abbreviation must be clarified in line 78, as follows, "Dissemination stations (DS)".

Response 4: The abbreviation has been clarified and used correctly after its citation (e.g., dissemination stations - DSs).

Comment 5: Line1: not in 2020?

Response 5: I believe this comment refers to line 81 of the manuscript. Manaus (Amazonas, Brazil) faced two major COVID-19 outbreaks: the first wave in April 2020, where despite the collapse of the healthcare system, surveillance continued with some gaps in data collection; and the second wave beginning in December 2020, with the spread of the P1 variant, first identified in Manaus. Due to the impact of this second wave and the collapse of the state's healthcare system, several isolation measures were taken to contain the spread of the virus, one of which was the temporary suspension of household visits for data collection during the period of social isolation.

Comment 6: Lines 100-107: It is not clear if the authors amplified a fragmento of the COI in orde to identify the mosquito species (if this is the case, please give the rpimers and PCR conditions). or if the authors search for mitochondrial SNPs in the reads with a program like mitoXXXX and then blaste these SNPs. Give a clearer explanation.

Response 6: We generated COI consensus using only the covered regions by the RadSeq and submitted the resulting subsequences to BLASTn.

Comment 7: Line 113: delete the stop after [18].

Response 7: The incorrect “.” has been deleted.

Comment 8: Line 117: replace "Population and structure genetics inferences" by "Population genetics structure inference" or "Genetic structure inference".

Response 8: The topic "Population and structure genetics inferences" has been corrected and replaced with "Population genetics structure inference."

Comment 9: Line 119: Provide the GGPlot2 package reference, as well as the R reference.

Response 9: We used ggplot2 version 3.5.0. This information will be included in both manuscript and references. Thank you.

Comment 10: Line 123: Ib. pvclust v2.2-0 package.

Response 10: This information will be included in both manuscript and references.

Comment 11: Line 136: Italicize "Ae. aegypti".

Response 11: Line 136: The term has been correctly italicized.

Comment 12: Lines 137-145: It seems like very poor sequencing depth and quality for both species. I was expecting something values ca. 20-30X. The same for quality (Phred score), ca. 35-40

Response 12: Thank you for your comment. We acknowledge that sequencing depth and quality obtained were below the expected values. However, this limitation is intrinsic to ddRADSeq technique when applied to large genomes such as those of the mentioned species (Aedes aegypti and Aedes albopictus). Despite this limitation, it is important to highlight that with the number of samples used in our study, we were able to recover valid and relevant genomic information for our research objectives. We appreciate your observation and are available to discuss any other questions.

Comment 13: Line 141: Italicize "Ae. albopictus ".

Response 13: Line 141: The term has been correctly italicized.

Comment 14: Line 147: Italicize "Ae. aegypti".

Response 14: Line 147: The term has been correctly italicized.

Comment 15: Lines 150-151: I would rather expect estimations of both Fis (deviation from HW proportions) and Fst (population structure).

Response 15: We estimated Fis using Plink with --het procedure, but the genotyping mean depth is too low for reliable individual estimates of Fis, so we used the Fst only, as the low depth bias is dilluted in the sample.

Comment 16: Line 155: Fst 0.015 does not indicate great differentiation. It would be interesting to estimate Fst within each neighborhood, to see is pesticide is a factor introducing (or erasing) signs of structuration. If data are pooled, the effect of pesticide cannot be distinguished.

Response 16: We appreciate and agree with your comment. Based on your suggestion, we have added a section to the manuscript discussion where we address the consideration that at a microgeographic scale and terms of the magnitude of the stratification signal for short distances, an FST value of 0.015 can be considered relatively high. This value suggests that there is more significant genetic structuring than expected at a local scale, and we discuss the possibility that factors such as pesticide use may be contributing to this differentiation.

Once again, we thank you for your observations to improve the quality of our study.

Comment 17: Italicize all scientific names across the whole manuscript and give the species authority.

Response 17: The entire text has been reviewed again to ensure the authority in the first citation of the species and species names consistently italicized throughout the manuscript.

Comment 18: Lines 192-194:  If both groups are in equilibrium, then how come that Tajima’s D are statistically different? Inf both are in equilibrium, then Tajima’s D for both groups do not differ statistically from 0. Thus, how both groups are different in this regard?

-Lines 1999-200: Is an Fst = 0.015 large enough using this kind of markers? Can you provide comparisons with other systems?

Response 18: We assumed that these populations are slightly different, as we observed significant estimate of Fst greater than zero. Anything significantly greater than zero for Fst estimate stands for HWE deviation, so we applied Tajima’s D for each strata. At this level of microgeographic distances, we think that even slight population differentiation must be considered.

Comment 19: Lines 205-207: I can see overlapping between both groups, although the Control group is far more diverse.

Response 19: Yes, that is correct. PCA reveals that most of the differentiation come from the different variability for each strata. We tested for PERMDISP with vegan::beta_betadisper() for R and the strata variability differs significantly between groups [F(1,50)=6.05, p=0.02*].

Comment 20: Lines 208-209: I think is the opposite pattern. Since the Control group is more diverse, then this should be the more variable group (I understand than the “intervention site” is the area of pesticide application, is that true?). The intervention (application of pesticide), must have exerted some selection for sure, lowering the genetic variation.

-Lines 209-211: that is true. Maybe I misunderstood the term “intervention site”.

Response 20: Yes, both statements are correct. In the first one, this is exactly what we expect to observe, a lower variability at the intervention site.

Comment 21: Lines 217-224: In other words, a selective pressure is acting in the intervention site, reducing genetic variation.

Response 21: Yes, that is correct. A sentence was added for better understanding of this paragraph.

Comment 22: Comparison with other studies should be useful. I suggest at least try to run a software like Structure or Admixture, in order to better reflect the lack of structure.

Response 22: We appreciate and agree with your suggestions. A paragraph comparing our results with those from other studies has been added to the discussion. However, we would like to clarify that Structure and Admixture are primarily designed to identify genetic structure and population composition by focusing on the presence of different genetic components. Since our focus is not on genetic structure but rather on the observed genetic variability, the use of these software tools may not be the most appropriate for the objectives of our study. In other words, the intervention seems to affect genetic diversity by reducing variability but did not alter the genetic composition of the population. 

Thank you and we are available for any further clarification.

Response to Comments on the Quality of English Language

Point 1: (x) English language fine. No issues detected.

Response 1: Thank you for considering our manuscript written in English to be of high quality and for your dedication in reviewing our manuscript.

Round 2

Reviewer 1 Report

Comments and Suggestions for Authors

Thank you to the authors for their efforts in providing more details to improve the clarity and significance of the manuscript. However, there are still two major concerns that have not been adequately addressed:

1.     Sample Size Justification: The rationale for selecting only 102 samples to represent the large populations from both control and experimental groups is still unclear. In their rebuttal letter, the authors mentioned that this sample size was chosen due to cost and funding constraints and that it is sufficient to represent the study areas. These arguments should be included in the main text, with a clear justification for why this number is adequate. Given there are 100 collection sites from both control and experimental groups combined, and the surveillance has lasted multiple years, the ideal approach would involve selecting at least one specimen per site, repeated annually. With multiple mosquito generations each year, genetic variation could accumulate sufficiently for testing yearly, necessitating at least 500 specimens to represent 100 sampling sites over, let’s say, five years. I suggest testing more specimens to enhance the study's robustness. If additional testing is not feasible, the authors should specify the sites and years from which the specimens were collected and acknowledge that their sample size may not fully represent the study areas across different sites and years.

2.     Importance of Studying Mosquito Genetic Structure: My previous comments requested clarification on why studying mosquito genetic structure is crucial for mosquito surveillance and vector-borne disease control. This information is highly relevant to public health departments, researchers, and readers and would greatly enhance the manuscript's significance and impact. Including this information in the introduction and discussion sections will strengthen the manuscript.

Author Response

Thank you to the authors for their efforts in providing more details to improve the clarity and significance of the manuscript. However, there are still two major concerns that have not been adequately addressed:

R: Dear Reviewer,

Thank you once again for your time and detailed analysis to improve our manuscript. We hope to address your questions and incorporate your suggestions appropriately in our revised manuscript.

  1. Sample Size Justification: The rationale for selecting only 102 samples to represent the large populations from both control and experimental groups is still unclear. In their rebuttal letter, the authors mentioned that this sample size was chosen due to cost and funding constraints and that it is sufficient to represent the study areas. These arguments should be included in the main text, with a clear justification for why this number is adequate. Given there are 100 collection sites from both control and experimental groups combined, and the surveillance has lasted multiple years, the ideal approach would involve selecting at least one specimen per site, repeated annually. With multiple mosquito generations each year, genetic variation could accumulate sufficiently for testing yearly, necessitating at least 500 specimens to represent 100 sampling sites over, let’s say, five years. I suggest testing more specimens to enhance the study's robustness. If additional testing is not feasible, the authors should specify the sites and years from which the specimens were collected and acknowledge that their sample size may not fully represent the study areas across different sites and years.

R: Thank you for your comment. We fully understand and agree with your viewpoint on the importance of a more robust experimental design involving a larger number of samples to better represent genetic variations over the years and across different collection sites. The initial approach would involve selecting 500 individuals, considering 50 collection sites from both control and experimental groups (totaling 100 combined points), in a spatio-temporal experimental design between 2017-2020. However, in the context of a Brazilian research group, we face severe limitations in investment and research resources, making it unfeasible to sequence the suggested number of samples (n). While we recognize that a larger sample size would increase the robustness of the study, the costs associated with sequencing at least 500 specimens are prohibitive within our current funding capabilities. Despite these limitations, we selected samples from the same collection points, considering the annual period (2017 to 2020) and during the rainy seasonal period when there is a higher availability of vectors, trying to be as geographically representative as possible. Additionally, one of our objectives was to observe the spatial dispersion of these mosquitoes based on the coefficient of relatedness between the collected individuals, which was not possible due to the number of SNPs recovered. It is also important to emphasize that, since the study of the application of the PPF intervention took place in a scenario of neighborhoods located in the center of a capital city, combined with control sites that have not been exposed to the intervention, the number of specimens collected was excessively large. Based on the mentioned collection points, the specimen selection for DNA extraction and analyses was made considering both sequencing cost and adequately representing both study areas (control and experimental). We believe that the carefully selected sample size, chosen to represent the study areas as best as possible, provides relevant and significant information for the scientific community, allowing us to observe genetic variation sufficiently in this spatio-temporal study. We appreciate your observations once again and have included a justification for this sample size in the manuscript text. We are committed to contributing to scientific advancement despite financial challenges and believe that our study offers valuable data for understanding the genetic structure of the mosquitoes in question.

  1. Importance of Studying Mosquito Genetic Structure: My previous comments requested clarification on why studying mosquito genetic structure is crucial for mosquito surveillance and vector-borne disease control. This information is highly relevant to public health departments, researchers, and readers and would greatly enhance the manuscript's significance and impact. Including this information in the introduction and discussion sections will strengthen the manuscript.

R: We sincerely appreciate these suggestions. We agree that this information is highly relevant for researchers and readers, so we have included these details in the introduction and discussion sections of the manuscript. We hope that these additions meet your expectations and contribute to the strengthening of our manuscript.
